Taxonomic review of the genus Stenocaris Sars (Copepoda, Harpacticoida, Cylindropsyllidae), with (re)descriptions of two Stenocaris species from the Far East

Kim Jong Guk 1 2
Cho Kyuhee 1
Yoon Seong Myeong 3 4 smyun@chosun.ac.kr
http://orcid.org/0000-0001-9004-8275 Lee Jimin 1 leejm@kiost.ac.kr
1 Marine Ecosystem Research Center, Korea Institute of Ocean Science & Technology , Busan , Republic of Korea
2 Division of Zoology, Honam National Institute of Biological Resources , Mokpo , Republic of Korea
3 Educational Research Group for Age-associated Disorder Control Technology, Chosun University , Gwangju , Republic of Korea
4 Department of Biology, College of Natural Sciences, Graduate School, Chosun University , Gwangju , Republic of Korea
Idris Izwandy
Electronic publication date: 2023 Jan 13
Publication date: 2023
Volume: 11
Electronic Location ID: e14623
Received 2022 Jul 27; Accepted 2022 Dec 2
Copyright: © 2023 Kim et al.
Copyright year: 2023
Copyright holder: Kim et al.
License: This is an open access article distributed under the terms of the Creative Commons Attribution License, which permits unrestricted use, distribution, reproduction and adaptation in any medium and for any purpose provided that it is properly attributed. For attribution, the original author(s), title, publication source (PeerJ) and either DOI or URL of the article must be cited.
License URL: https://creativecommons.org/licenses/by/4.0/

Keywords: Huysicaris gen. nov.; Huysicaris arenicola (Wilson, 1932) comb. nov.; Huysicaris kliei (Kunz, 1938) comb. nov.; interstitial meiofauna; Korean waters; Russian waters; Stenocaris figroloensis sp. nov.; Stenocaris marcida sp. nov.

Funding: National Marine Biodiversity Institute of Korea 2022M01100 National Institute of Biological Resources NIBR201501201 Korea Institute of Ocean Science and Technology PEA0016 This work was supported by the National Marine Biodiversity Institute of Korea (No. 2022M01100), the National Institute of Biological Resources (No. NIBR201501201), and the research programme (No. PEA0016) of the Korea Institute of Ocean Science and Technology. The funders had no role in study design, data collection and analysis, decision to publish, or preparation of the manuscript.

==============================
The taxonomic concept of the genus Stenocaris Sars, 1909 is uncertain because none of the synapomorphies for the species of Stenocaris are defined. Detailed comparison of previous records of Stenocaris minor (T. Scott, 1892) from different localities reveals that this species represents a species complex composed of two species, S. minor s. str. and S. minor sensu Cottarelli & Venanzetti, 1989. Because the latter species has fundamental differences in the nature of the fifth leg in females and the sexual dimorphism of the second leg in males, we propose a new species for S. minor sensu Cottarelli & Venanzetti, 1989, S. figaroloensis sp. nov. We also suggest that S. minor sensu Apostolov, 1971, S. minor sensu Marinov, 1971, and S. minor sensu Apostolov & Marinov, 1988 from the Black Sea and S. minor sensu Wilson, 1932 from North America should be relegated to species inquirenda in the genus. Taxonomic review of the morphology of all Stenocaris species indicated that the generic concept must be restricted to accommodate S. minor s. str., S. gracilis Sars, 1909, S. intermedia Itô, 1972, S. figaroloensis sp. nov., and the South Korean new species, S. marcida sp. nov., based on the synapomorphic condition of the confluent fifth leg in males. As a result of our analysis, two Stenocaris species, S. baltica Arlt, 1983 and S. pygmaea Noodt, 1955, are transferred to the genus Vermicaris Kornev & Chertoprud, 2008 as V. baltica (Arlt, 1983) comb. nov. and V. pygmaea (Noodt, 1955) comb. nov. based on the synapomorphic characters of a reduced condition of the second and fifth legs. Additionally, S. arenicola Wilson, 1932 and S. kliei (Kunz, 1938) are allocated to a new genus, Huysicaris gen. nov., mainly characterized by obvious caudal rami with a recurved dorsal spinous process and convex inner margins, as H. arenicola (Wilson, 1932) comb. nov. and H. kliei (Kunz, 1938) comb. nov. A marine interstitial harpacticoid collected from the subtidal substrate off Dok-do Island in the East Sea of South Korea is proposed as S. marcida sp. nov. and the distribution of S. intermedia, originally known from its type locality in Japanese waters only, is extended to the East Sea of Korea and Russia. We provide their detailed descriptions and illustrations and discuss the morphological characters supporting their identities.

Introduction

Harpacticoids are the second-most species-rich and abundant group in the meiofaunal community after nematodes (Hicks & Coull, 1983). They exhibit diverse morphological adaptations as a result of the benthic habitats they have colonized, e.g., endobenthic species have specialized thoracic appendages or a well-developed rostrum to facilitate burrowing (Gee & Huys, 1996; Corgosinho, 2012; Kim & Lee, 2020), and interstitial species underwent three morphological trends to swim or crawl within the lacunae between sediment particles: miniaturization of body size, elongation along the body axis, and reduction of the setal armature of the thoracic legs (Noodt, 1971; Martínez Arbizu & Moura, 1994; Huys & Conroy-Dalton, 2006a; Corgosinho, 2012).

The family Cylindropsyllidae Sars, 1909, currently comprising 39 species in 13 genera, is a representative of mesopsammic harpacticoids (Richter, 2019). Members of this family have an interstitial lifestyle and exhibit the above morphological trends except for miniaturization, and a cylindrical or vermiform habitus without a distinct constriction between the prosome and urosome, and slender swimming legs with reduced segmentation and setal armature (Huys & Conroy-Dalton, 2006a; Richter, 2019). These morphological features likely enhance their flexibility and ability to wriggle between sand grains and shell gravel in shallow coastal and sublittoral environments (Huys, 1988; Huys & Conroy-Dalton, 1993, 2006a; Huys & Lee, 2018), but some species have been documented in deep-sea mud (Becker, Noodt & Schriever, 1979; Moura & Pottek, 1998; Richter, 2019). Huys (1992) reported a strong correlation between the distribution of cylindropsyllid species and sediment conditions. Cylindropsyllids are also morphologically characterized by the fifth leg forming a single plate in both sexes and the sexual dimorphism of the second and third swimming legs in the male (Huys, 1988).

Since Lang’s (1948) familial concept of the Cylindropsyllidae (a subfamily at that time), the genus Stenocaris Sars, 1909, had until the 1970s served as a repository to accommodate more advanced cylindropsyllid harpacticoids (i.e., retaining a slender-type maxilliped with a geniculate claw, a non-prehensile endopod of the first leg, the second and third legs with reduced endopod segmentation), without any morphologic or phylogenetic inference. Therefore, the monophyly of this genus is questionable (Huys, 1988; Huys & Conroy-Dalton, 1993, 2006b) and apomorphies have not been defined (Richter, 2019). These taxonomical problems were partially resolved by the following contributions. Apostolov (1982) established a new genus Stenocaropsis Apostolov, 1982 to accommodate Stenocaris pristina Wells, 1968 and Stenocaris valkanovi Marinov, 1974 which possess two-segmented endopods on the second and third legs against Sars’s (1909) generic diagnosis with the one-segmented endopods on these legs. However, Apostolov (1982) questionably retained in Stenocaris two deep-sea species, Stenocaris abyssalis Becker, Noodt & Schriever, 1979 and Stenocaris profundus Becker, Noodt & Schriever, 1979, which have the same endopodal segmentation as Stenocaropsis. Although Kunz (1994) assigned the latter two deep-sea species to Stenocaropsis, Moura & Pottek (1998) established a new genus Selenopsyllus Moura & Pottek, 1998 for Becker, Noodt & Schriever’s (1979) two deep-sea species and two new Antarctic species, Selenopsyllus dahmsi Moura & Pottek, 1998 and Selenopsyllus antarcticus Moura & Pottek, 1998. The latter were characterized by a six-segmented female antennule, at most, with an aesthetasc on the third segment (Moura & Pottek, 1998). Huys & Conroy-Dalton (1993) transferred Stenocaris kerguelenensis Bodiou, 1977 to a new monophyletic genus Navalonia Huys & Conroy-Dalton, 1993. Kornev & Chertoprud (2008) removed Stenocaris minuta Nicholls, 1935 and Stenocaris pontica Chappuis & Serban, 1953, and placed them in a new genus, Vermicaris Kornev & Chertoprud, 2008. As a result of these efforts, Stenocaris currently comprises seven valid species: Stenocaris minor (T. Scott, 1892), S. gracilis Sars, 1909 (type species; cf. Huys, 2009: 98), S. arenicola Wilson, 1932, S. kliei Kunz, 1938, S. pygmaea Noodt, 1955 (Apostolov (1972) and Apostolov & Marinov (1988) considered it a junior synonym of V. pontica (Chappuis & Serban, 1953), but it was reinstated as a valid species of Stenocaris by Huys & Conroy-Dalton (1993: 295)), S. intermedia Itô, 1972, and S. baltica Arlt, 1983. However, their phylogenetic relationship is still unclear.

Taxonomical surveys on the marine harpacticoid family Cylindropsyllidae from South Korea have been limited. Back et al. (2009) only reported the occurrence of this family in a sublittoral environment off Sungap-do Island in the Yellow Sea during a survey of the meiofaunal community in a sand-mining area. The following sections provide a taxonomical report of the family in this area based on the specimens identified as S. intermedia and a new Stenocaris species from the subtidal sediments of the East Sea (including Russian waters), describe the morphological characteristics supporting their identities, and review the phylogenetic relationships among Stenocaris species to resolve the polyphyly of the genus.

Materials and Methods

Sediment samples were collected from the sublittoral benthic environments off Ulleung-do and Dok-do Islands (South Korea) and the coast of Primorsky (Russia) in the East Sea (Sea of Japan) via scuba diving (depths of 8.0–30.5 m) or with a Smith-McIntyre grab (depth of 73.3 m) (Fig. 1). These field samplings were approved by the National Marine Biodiversity Institute of Korea (MABIK) and the National Institute of Biological Resources (NIBR). Samples were immediately preserved in 95% ethanol or in 4% formalin solution. In the laboratory, they were filtered through a 50-μm sieve under tap water. Harpacticoid copepods were removed using a Pasteur pipette under a Leica M165 C stereomicroscope. Specimens of two Stenocaris species were prepared and mounted in lactic acid on a temporary reverse slide (Humes & Gooding, 1964). Total body length was measured from the anterior tip of the rostrum to the posterior end of the caudal rami in lateral view using an Olympus DP28 microscope camera. Line drawings of the whole specimen and its dissected appendages were prepared using a camera lucida on an Olympus BX53 compound microscope equipped with a differential interference contrast objective. After morphological examination, the dissected parts were transferred to and permanently mounted in glycerin on Higgins-Shirayama (H-S) slides comprising two coverslips (Shirayama, Kaku & Higgins, 1993). The slides were deposited in the MABIK or NIBR in South Korea. Maps of sampling stations and the distribution of Stenocaris species were generated using Ocean Data View ver. 5.6.1 (Schlitzer, 2022).

Figure 1 Map of the sampling stations of Stenocaris marcida sp. nov. (filled triangle, ▲) and S. intermedia Itô, 1972 (filled circle, ●).

We followed Huys et al. (1996) for the morphological terminology of the body and appendages, and Lang (1934) for the setal armature formulae of swimming legs. Abbreviations used in the text, tables and figure captions are as follows: acro, acrothek composed of one aesthetasc and two setae (fused basally); ae, aesthetasc; apo, apophysis; EXP(ENP)1(2, 3), first (second, third) exopodal (endopodal) segment; P1–P6, first to sixth thoracic leg. We also followed Huys & Conroy-Dalton’s (2006b) designations for armature elements on the fifth leg (innermost = seta a, second innermost = seta b … outermost = seta h in females and seta f in males) to compare their positions and modifications in other genera or species.

The electronic version of this article in Portable Document Format (PDF) will represent a published work according to the International Commission on Zoological Nomenclature (ICZN), and hence the new names contained in the electronic version are effectively published under that Code from the electronic edition alone. This published work and the nomenclatural acts it contains have been registered in ZooBank, the online registration system for the ICZN. The ZooBank LSIDs (Life Science Identifiers) can be resolved and the associated information viewed through any standard web browser by appending the LSID to the prefix http://zoobank.org/. The LSID for this publication is: (urn:lsid:zoobank.org:pub: 9F8C21FE-74C1-4F7C-BA5C-22B9CC0869DB). The online version of this work is archived and available from the following digital repositories: PeerJ, PubMed Central and CLOCKSS.

Results

Taxonomy

Order Harpacticoida Sars, 1903

Family Cylindropsyllidae Sars, 1909

Genus Stenocaris Sars, 1909

Type species.—Stenocaris gracilis Sars, 1909

Other species.—S. minor (T. Scott, 1892), S. intermedia Itô, 1972, S. figaroloensis sp. nov., and S. marcida sp. nov.

Amended diagnosis.—Cylindropsyllidae. Habitus slender, cylindrical. Rostrum triangular, as long as first antennular segment, defined at base. Genital somite and third urosomite completely fused in ♀ forming genital double-somite. Anal somite as long as penultimate somite. Caudal rami elongate, at least 2.5 times as long as wide; seta I diminutive, principal seta V styliform or composite (styliform proximal part and slender setiform distal part) in ♀. Antennule six- or seven-segmented, with aesthetasc on fourth and terminal segments in ♀, haplocer and nine- or ten-segmented in ♂; second segment distinctly elongated in both sexes. Antenna with allobasis; exopod one-segmented, bearing two terminal setae. Mandibular palp two-segmented, uniramous, with one seta on basis, and one lateral and three or four distal setae on endopod. Maxillule with two setae on coxal endite; exopod and endopod represented by two setae and three or four setae, respectively. Maxilla with two syncoxal endites and one-segmented endopod. Maxilliped well-developed, with geniculate endopodal claw. P1 endopod non-prehensile; ENP2 with one small and two long setae. P2–P4 exopods three-segmented; P2–P3 EXP3 with two outer spines and P4 EXP3 with one or two outer spines; P2 EXP3 without and P3–P4 EXP3 with one inner seta; P2 EXP3 in ♂ strongly modified, as long as EXP1 and EXP2 combined, with wavy inner margin, and one stout and distally recurved apical spine, and lacking inner element; P4 exopod longer than those of P2–P3. Endopods in ♀ one-segmented in P2–P3 and two-segmented in P4. P2 in ♂ with spinous anterior process on basis and two-segmented endopod. P3 in ♂ with two-segmented endopod; ENP1 with apophysis in S. minor and S. figaroloensis sp. nov., but without apophysis in other species; ENP2 modified into a stout apophysis in S. gracilis and S. intermedia, subovate in S. minor and S. figaroloensis sp. nov., and with spinose projection in S. marcida sp. nov. Setal armature formula of P1–P4 as in Table 1.

Table 1 Armature formula of P1–P4 in the genus Stenocaris Sars, 1909.

	Exopod	Endopod	
P1	0.0.112	1.120	
P2	0.0.022 [0.0.022 in ♂]	11–20 [1.010 in ♂]	
P3	0.0.122	01–20 [0.apo10 or apo.010 in ♂]	
P4	0.0.121–2	0.0–110	

P5 forms a single plate in both sexes, with seven distal elements in ♀, of which innermost one spine-like (fused basally in S. gracilis, S. intermedia and S. marcida sp. nov.) and second innermost one spine-like (seta-like in S. minor and S. figaroloensis sp. nov.), and five distal setae in ♂; baseoendopods separate in ♀, but fused medially in ♂.

P6 with two or three setae in ♀ and three setae in ♂.

Remarks.—See the discussion (below) on a Stenocaris minor species complex, and taxonomic positions of S. arenicola, S. baltica, S. kliei, and S. pygmaea.

Stenocaris marcida sp. nov.

urn:lsid:zoobank.org:act: 45861574-3D8E-42CB-ADEE-E32EA4F3334D

Figures 2–7

Figure 2 Stenocaris marcida sp. nov., female, holotype, MABIK CR00252790 (A–C) and paratype, MABIK CR00252792 (D).

(A) Habitus, dorsal; (B) Habitus, lateral; (C) Urosome, ventral; (D) Caudal ramus, lateral. Scale bars are given in µm.

Figure 3 Stenocaris marcida sp. nov., female, holotype, MABIK CR00252790 (A, B).

(A) Rostrum and antennule; (B) P1. Scale bars are given in µm.

Figure 4 Stenocaris marcida sp. nov., female, holotype, MABIK CR00252790 (A–F) and paratype, MABIK CR00252792 (G).

(A) Antenna; (B) Mandible; (C) Maxillule; (D) Maxilla; (E) Maxilliped; (F) P5; (G) Genital field. Scale bars are given in µm.

Figure 5 Stenocaris marcida sp. nov., female, holotype, MABIK CR00252790 (A–C), male, allotype, MABIK CR00252791 (D).

(A) P2; (B) P3; (C) P4; (D) P5. Scale bars are given in µm.

Figure 6 Stenocaris marcida sp. nov., male, allotype, MABIK CR00252791.

(A) Habitus, dorsal; (B) Antennule. Scale bars are given in µm.

Figure 7 Stenocaris marcida sp. nov., male, allotype, MABIK CR00252791.

(A) Urosome, ventral; (B) P2; (C) P3. Scale bars are given in µm.

Type locality.—Sublittoral sandy sediments off Dok-do Island, Korea (37°14′5.63″N 131°51′43.19″E); 73.3 m depth.

Type material.—Holotype: ♀ dissected on 11 slides (MABIK CR00252790) collected from the type locality, November 2, 2018. Allotype: ♂ dissected on eight slides (MABIK CR00252791), collection data as in holotype. Paratypes: 1♀ (MABIK CR00252792), 2♀♀, 2♂♂ (MABIK CR00252793) preserved together in a vial with 99% ethanol, collection data as in holotype; J.G. Kim leg.

Description of female (based on the holotype MABIK CR00252790).—Total body length about 736.0 µm (length range = 736.0–842.8 µm; mean = 786.7 µm, n = 4); habitus (Figs. 2A and 2B) slender, cylindrical, without distinct separation between prosome and urosome. Integument of cephalothorax, all somites, and caudal rami pitted, with several sensilla and pores except for penultimate somite. Posterior border of cephalothorax and all somites with smooth hyaline frill. Cephalothorax occupying 25% of body length, slightly tapering anteriorly in dorsal aspect. P5-bearing somite longer than third free pedigerous somite. Genital somite and third urosomite completely fused forming genital double-somite, longer than preceding somite; genital pores (Fig. 4G) separate, each covered by small plate bearing two setae (representing P6); mid-ventral copulatory pore located at same level of genital pores, probably covered by membrane with opening (Fig. 4G). Two free urosomites and anal somite slightly tapering posteriorly. Anal somite elongate, as long as penultimate somite, with one pair of dorsal sensilla and three pairs of ventrolateral pores; operculum wide, with smooth posterior margin.

Rostrum (Figs. 2A and 3A) prominent, triangular, defined at base, not exceeding first antennulary segment, slightly longer than maximum width; with one pair of subapical sensilla.

Caudal rami (Figs. 2A–2D) slightly divergent, as long as anal somite, about 3.1 times as long as maximum width; medial margin slightly concave in dorsal view; dorsal surface with one pore subdistally, and ventral surface with one pore proximally and one pore subdistally; with seven setae: setae I and II inserted in distal fifth of lateral margin at same level; seta I diminutive, noticeable at high magnification; seta II as long as caudal ramus; seta III arising from dorsal surface subdistally, about 1.3 times longer than caudal ramus; terminal setae IV and V fused basally; seta IV slender, about 1.5 times as long as caudal ramus; principal seta V composite, with styliform proximal part slightly shorter than seta IV, and long slender setiform distal part; seta VI short, ventrally inserted in distal margin; seta VII tri-articulate basally, arising in middle of caudal ramus on dorsal surface, slightly shorter than caudal ramus.

Antennule (Fig. 3A) slender, longer than cephalothorax, seven-segmented. First segment small, with one small seta ventrally. Second segment elongated, about 5.9 times as long as preceding one, with four bi-articulate and four slender setae, and one tube pore. Third segment small, twice as long as wide, with four bi-articulate setae. Fourth segment smaller than preceding one, with one seta and one distal peduncle bearing one aesthetasc and one small seta. Fifth segment smaller than preceding one, with one seta. Sixth segment smallest, with one seta. Terminal segment slightly longer than two preceding ones combined, tapering distally, with one seta, six bi-articulate setae, and acrothek. Setal formula: 1-[1], 2-[8], 3-[4], 4-[1 + (1 + ae)], 5-[1], 6-[1], 7-[7 + acro].

Antenna (Fig. 4A). Coxa small, unornamented. Allobasis elongate, 3.8 times as long as maximum width. Exopod arising at proximal third of allobasis, one-segmented, slender, with two long setae distally. Free endopodal segment elongate; inner margin with two unipinnate spines, one proximal spinule, and one subdistal group of spinules; distal margin oblique, with one delicate seta (indicated by arrowhead in Fig. 4A), two pinnate spines, and three geniculate setae, of which outermost one spinulose and fused to one delicate seta; with hyaline frills distally and subdistally.

Mandible (Fig. 4B). Coxa well-developed, transversely elongate, with one medial bulge; gnathobase with one bicuspidate and three multicuspidate teeth, and one unipinnate seta fused to one stout spine. Palp uniramous, two-segmented; basis elongate, about 4.6 times as long as wide, with one subdistal seta; endopod elongate, twice as long as wide, with one lateral seta, and two distal sets of two setae basally fused.

Maxillule (Fig. 4C). Praecoxa with one row of posterior spinules; arthrite well-developed, with seven spines distally and two pinnate setae subdistally, and one long and one small seta on anterior surface. Coxa small, unornamented; cylindrical endite with one spinulose and one bare seta. Basis with one row of spinules both posteriorly and anteriorly; distal margin with one spinulose and three bare setae; subdistal endite with two bare setae. Exopod and endopod represented by two and three bare setae, respectively.

Maxilla (Fig. 4D). Syncoxa unornamented, with two endites: proximal endite with one unipinnate spine, and one bare and one unipinnate seta distally; distal endite with three unipinnate setae. Allobasis drawn out into stout claw, with spinules subdistally and two setae proximally. Endopod one-segmented, with four distal setae.

Maxilliped (Fig. 4E). Syncoxa elongate, 2.7 times as long as wide, subdistally with one unispinulose seta. Basis longer than preceding segment, 2.8 times longer than maximum width, with one outer spinule subdistally. Endopod small, with one unipinnate and geniculate claw.

P1 (Fig. 3B). Praecoxa large, triangular, unornamented. Coxa rectangular, about 0.6 times as long as broad, pitted, with one row of anterior spinules; posterior surface with three rows of spinules. Basis smaller than preceding segment, with one row of antero-distal spinules, one anterior pore, and one long inner seta (uniplumose distally); outer seta absent. Exopod three-segmented, slightly longer than endopod, with spinular ornamentation along outer margins; length ratio of EXP1–EXP3 1: 0.81: 0.78; EXP1 and EXP2 with one small unipinnate outer spine; EXP3 with four unispinulose spines, of which distal and inner ones geniculate. Endopod reaching middle of EXP3, two-segmented, with outer spinular ornamentation; ENP1 elongate, slightly exceeding distal end of EXP1, with one unipinnate seta posteriorly; ENP2 slender, as long as preceding segment, distally with one unipinnate geniculate spine, and one long geniculate seta (uniplumose proximally), and posteriorly with one small bare seta.

P2 (Fig. 5A). Praecoxa small, unornamented. Intercoxal sclerite subrectangular, distal margin concave. Coxa rectangular, pitted (not figured), unornamented. Basis smaller than preceding segment, with one bare outer seta and one anterior pore. Exopod three-segmented, much longer than endopod; length ratio of EXP1–EXP3 1: 0.66: 1.05; EXP1 and EXP2 with spinular ornamentation along outer margin subdistally and with distal surface frill, spinous process bearing anterior spinules distally, and unipinnate outer spine, respectively; EXP3 with spinular ornamentation along outer and distal margins, two unipinnate outer spines, and two pinnate distal spines, of which inner one ornamented with more distinct spinules. Endopod one-segmented, reaching distal fourth of EXP1, 3.6 times as long as broad, with one pinnate spine distally, one small bare posterior seta subdistally, and one long uniserrate posterior seta proximally.

P3 (Fig. 5B). Protopod and intercoxal sclerite as in P2, but smaller than in P2; outer seta of basis longer than those of P2 and P4. Ornamentation and segmentation of both rami largely as in P2. Length ratio of EXP1–EXP3 1: 0.66: 1.19; exopodal armature as in P2 except for presence of one uniserrate inner seta in EXP3. Endopod one-segmented, not exceeding middle of EXP1, with one stout pinnate spine distally and one small posterior seta subdistally.

P4 (Fig. 5C). Protopod and intercoxal sclerite as in P2 and P3, but smaller than in P2 and P3. Exopod longer than in P2 and P3; length ratio of EXP1–EXP3 1: 0.97: 0.65; EXP1 with three groups of outer spinules, outer distal corner not produced, with one row of stout spinules, and distal hyaline frill expanded roundly; ornamentation of EXP2 as in EXP1 except for presence of only one group of outer spinules; EXP1 and EXP2 with one pinnate outer spine; EXP3 with distal hyaline frill, two pinnate outer spines, two pinnate distal spines, and one uniserrate inner seta. Endopod two-segmented, as long as EXP1; ENP1 without setal armature; ENP2 smaller and slenderer than preceding segment, with one fishbone-like spine distally and one small posterior seta subdistally.

P5 (Fig. 4F). Exopod and baseoendopod fused forming a single plate; with eight elements: seta a spiniform, fused to baseoendopod, serrate subdistally; other setae setiform, setae b, d, e longer than others, and seta c half-length of seta b, and seta f shortest; outer basal seta h very long, bi-articulate basally, uniplumose distally.

Description of male (based on allotype MABIK CR00252791). Body slightly shorter than female, about 702.0 µm (range = 698.5–744.1 µm; mean = 714.9 µm, n = 3); habitus (Fig. 6A) as in female, except for the following characters.

Cephalothorax (Fig. 6A) with mid-dorsal integumental window.

Urosome (Figs. 6A and 7A) six-segmented, genital somite and third urosomite separated; spermatophore elongate, about 1/5 of body length, extending to the middle of third urosomite.

Caudal rami (Figs. 6A and 7A). Seta III 1.8 times as long as caudal ramus. Seta V composite as in female, but with more slender styliform proximal part.

Antennule (Fig. 6B) haplocer, ten-segmented, with geniculation between seventh and eighth segments. First segment small, posteriorly with one patch of minute spinules, one row of spinules, and one bare seta. Second segment longest, with one tube pore, one uniplumose, one bi-articulate, and seven bare setae. Third segment small, being at right angle with previous segment, with one bi-articulate and six bare setae; outer margin concave. Fourth segment smaller than preceding one, with two setae. Fifth segment swollen, with one uniplumose, one bi-articulate and three single setae, and one distal peduncle bearing one long aesthetasc and one bare seta (fused basally). Sixth segment small, with one long and one small uniplumose seta. Seventh segment elongate, slightly curved inwardly, with one uniplumose and two bare setae. Eighth segment slightly shorter than preceding one, with one minute seta. Ninth segment smallest, unarmed. Terminal segment as long as seventh segment, tapering distally, with one bare and six bi-articulate setae, and acrothek. Setal formula: 1-[1], 2-[9], 3-[7], 4-[2], 5-[5 + (1 + ae)], 6-[2], 7-[3], 8-[1], 9-[0], 10-[7 + acro].

P2 (Fig. 7B). Protopod and intercoxal sclerite as in female. Basis with spinose inner process anteriorly. EXP1 and EXP2 as in female. EXP3 elongate, 0.85 times as long as EXP1 and EXP2 combined, uneven along inner margin, with three pinnate outer spines and one modified stout distal seta recurved distally, and with one oblique row of minute spinules. Endopod two-segmented, not reaching middle of EXP2; ENP1 with convex inner margin and one distally uniserrate seta posteriorly; ENP2 1.7 times as long as preceding segment, with one small, distally recurved distal seta.

P3 (Fig. 7C). Protopod and intercoxal sclerite as in female. Basis with one row of anterior spinules proximally and distally. Exopod as in female. Endopod two-segmented, not reaching distal end of EXP1; ENP1 small, slightly longer than wide, unarmed; ENP2 about 1.5 times as long as preceding segment, with one small and stout distal seta, and one spinose projection on inner margin.

P5 (Fig. 5D). Baseoendopods of right and left legs fused medially; each with one lateral and three anterior pores. Exopod and baseoendopod fused forming a single plate, tapering distally, with truncate distal end; with six elements: seta a smallest, seta b twice as long as rami length, seta c slightly shorter than seta b, setae d and e slightly longer than seta a, and outer basal seta f three times as long as ramus, bi-articulate basally, uniplumose distally.

P6 (Fig. 7A) asymmetrical, only one functional, each represented by small plate, with one bi-articulate and two bare setae, of which middle one longest.

Variability.—Proximal styliform part of composite caudal seta V broken off in several females (see Fig. 2D). This could lead to the misconception that the female’s caudal seta V is styliform.

Etymology.—The specific name marcida is derived from the Latin adjective marcidus, meaning withered, and alludes to the relatively short seta on the male P3 ENP2. It is in the nominative singular, gender feminine.

Remarks.—Stenocaris marcida sp. nov. resembles Scott’s (1892) S. minor s. str. in the seven-segmented female antennule and armature of swimming legs, and in the setiform seta b (= second innermost seta) on female P5. However, the new species is characterized by the following structures: (1) seta a (= innermost spine) on female P5 is fused to segment, and distally serrate and blunt (not fused to ramus and pinnate in S. minor s. str.); (2) the recurved, stout distal spine of male P2 EXP3 is slightly shorter than the length of P2 EXP1 (very long, distinctly longer than P2 EXP1 and EXP2 combined in S. minor s. str.); (3) male P2 EXP3 approximately twice as long as EXP2 (slightly longer than EXP2 in S. minor s. str.); (4) male P2 ENP2 approximately twice as long as ENP1, bearing a small distal spine, which is setule-like in the distal part (slightly longer than ENP1, with a long and pinnate seta in S. minor s. str.); (5) the sexual dimorphism of male P3 is different in that male P3 ENP2 of S. marcida sp. nov. exhibits an apophysis of a small spinose process and has a spine-like seta shorter than the distal segment, while, in S. minor s. str., a stout apophysis appears in P3 ENP1, and subovate ENP2 has a long and slender distal seta; (6) caudal seta V is composite, and it is composed of a styliform proximal part and a slender setiform distal part (it is most probably non-composite in both sexes in S. minor s. str.). In addition, caudal seta VI in both sexes of S. minor s. str. (Scott, 1892, 1900) is very small and bulbous whereas it is normal in the new species.

The new species shares sexual dimorphism of P3 with S. gracilis and S. intermedia, suggesting a close relationship. However, S. marcida sp. nov. is readily distinguishable from them by the combination of the female seven-segmented antennule (vs. six-segmented in the other two species), male P3 ENP2 with a small spinous process (vs. absent, but the segment drawn out into an apophysis in the other two species), and the setiform seta b on female P5 (vs. spiniform in the other two species; Sars, 1909; Itô, 1972).

Stenocaris intermedia Itô, 1972

Figures 8–12.

Figure 8 Stenocaris intermedia Itô, 1972, female (A, B) and male (C) from Korean waters.

(A) Habitus, dorsal; (B) Urosome, ventral; (C) Habitus, dorsal. Scale bars are given in µm.

Figure 9 Stenocaris intermedia Itô, 1972, female (A, B) and male (C) from Korean waters.

(A) Antennule; (B) Antenna; (C) Antennule. Scale bars are given in µm.

Figure 10 Stenocaris intermedia Itô, 1972, female from Korean waters.

(A) Mandible; (B) Mandibular ganathobase; (C) Maxillule; (D) Maxilla; (E) Maxilliped; (F) P5; (G) Genital field, ventral; (H) Genital field, lateral. Scale bars are given in µm.

Figure 11 Stenocaris intermedia Itô, 1972, females from Korean (A–D) and Russian (E) waters.

(A) P2; (B) P3; (C) P4; (D) P5; (E) P5. Arrowheads indicate delicate and setule-like elements. Scale bars are given in µm.

Figure 12 Stenocaris intermedia Itô, 1972, male from Korean waters.

(A) Urosome, ventral; (B) Caudal ramus, lateral; (C) P2; (D) P3; (E) P5; (F) P6. Scale bars are given in µm.

Stenocaris intermedia Itô, 1972: p. 323, Figs. 13–16; Kornev & Chertoprud, 2008: p. 280, Fig. 146E.

Material examined: 2♀♀, 1♂ (NIBRIV0000900838, NIBRIV0000900839, NIBRIV0000900840) each dissected on a slide, 5♀♀, 4♂♂ (NIBRIV0000900841) preserved together in a vial with 99% ethanol, sublittoral sediments at Gwaneum-do Islet off Ulleung-do Island in the East Sea of Korea (37°33′02.60″N 130°54′26.50″E), September 16, 2016, 30.5 m depth, T.W. Jung leg.; 1♀, 1♂ (NIBRIV0000900842, NIBRIV0000900843) each dissected on a slide, 6♀♀, 4♂♂ (NIBRIV0000900844) preserved together in a vial with 99% ethanol, sublittoral sediments at the coast of Primorsky in the East Sea of Russia (42°37′20.00″N 131°07′41.40″E), October 9, 2019, water depth 8 m, S.H. Kim, J.W. Kang leg.

Description of female (based on NIBRIV0000900838).—Total body length ranged 966.1–1087.3 µm (mean = 1016.1 µm, n = 7; Korean population); body (Fig. 8A) slender, cylindrical, without clear distinction between prosome and urosome; integuments pitted and covered with sensilla and pores except for penultimate somite; smooth hyaline frill present in all somites. Cephalothorax representing 18% of body length. Free pedigerous somites slightly narrower than cephalothorax. First urosomite (P5-bearing somite) slightly tapering towards anterior end and other urosomites slightly tapering towards posterior end. Genital somite and third urosomite completely fused in female forming genital double-somite (Fig. 8A); genital slits separate, each covered by small plate with three small setae (representing P6) and copulatory pore covered by oval-shaped bulb (Figs. 10G and 10H). Anal somite 1.4 times as long as wide, with one pair of lateral secretory pores subdistally (Fig. 8B) and one pair of dorsal sensilla; anal operculum wide, moderately rounded, unarmed (Fig. 8A).

Rostrum (Fig. 8A) triangular, defined at base, with one pair of subapical sensilla; slightly shorter than maximum width.

Caudal rami (Figs. 8A and 8B) slightly divergent, cylindrical, about three times as long as maximum width, with one dorsal and two lateral pores; with seven setae: position of each seta as in S. marcida sp. nov.; seta I diminutive, shortest; seta II short, about 1/3 of ramus length; seta III displaced to dorsal surface subdistally, as long as seta II; seta IV slightly shorter than setae II and III, fused to seta V basally; principal seta V conical, spiniform, about 1/3 of ramus length; seta VI short, as long as setae II and III; seta VII tri-articulate at base, dorsally inserted near inner margin in middle of ramus, as long as setae II and III.

Antennule (Fig. 9A) six-segmented. First segment small, with one small seta, and ornamented with one row of outer spinules and two rows of minute posterior spinules. Second segment longest, about 3.4 times as long as maximum width, with one secretory pore and eight setae (four bare and four bi-articulated). Third segment 0.4 times shorter than preceding one, with four setae, of which two setae bi-articulated basally. Fourth segment smaller then preceding one, about 1.7 times as long as wide, with one long seta and one distal peduncle bearing one seta and one aesthetasc (basally fused). Fifth segment shortest, about 1.8 times as long as wide, with one bi-articulated seta. Sixth segment slightly tapering towards apical end, about 5.5 times as long as maximum width, with two bare setae, six bi-articulate setae, and acrothek. Armature formula: 1-[1], 2-[8], 3-[4], 4-[1 + (1 + ae)], 5-[1], 6-[8 + acro].

Antenna (Fig. 9B). Coxa small, with one row of minute spinules. Allobasis elongate, about four times as long as wide, with two rows of spinules near abexopodal margin. Exopod slender, one-segmented, with one pinnate and one plumose seta. Free endopodal segment, with one proximal row of inner spinules, one subdistal row of anterior spinules, one subdistal row of posterior spinules, and one distal outer hyaline frill; lateral armature composed of two uniserrate inner spines; distal armature comprised of one minute bare seta (indicated by arrowhead in Fig. 9B), two serrate spines, three unipinnate, geniculate setae, of which outermost one bearing row of stout spinules and basally fused to adjacent small seta.

Mandible (Figs. 10A and 10B) with well-developed coxa bearing one medial bulge and one row of spinules; gnathobase well-developed, with one bi-cuspidate and three multicuspidate teeth, one unipinnate seta fused to small spine, and one row of minute spinules. Uniramous palp consisting of basis and one-segmented endopod; basis elongate, about 3.7 times as long as maximum width, with one subdistal seta; endopod small, about 2.3 times as long as wide, with one lateral and four apical setae (two sets of setae merged basally).

Maxillule (Fig. 10C). Praecoxa large, with one row of posterior spinules; well-developed arthrite with seven spines distally, of which two anterior ones ornamented with one large spinule subdistally, and one anterior spine ornamented with few long spinules proximally, two plumose setae laterally, and one long and one small bare seta on anterior surface; posterior surface with two rows of minute spinules, and lateral margin with one spinule. Coxal endite reaching middle of praecoxal arthrite, with one pinnate spine and one bare seta. Basis with one row of spinules anteriorly, and one pinnate and three bare setae distally; subdistal endite with two bare setae. Both exopod and endopod incorporated into basis, represented by two and three setae, respectively.

Maxilla (Fig. 10D). Syncoxa large, with two endites: proximal endite distally with one unipinnate spine (fused to endite basally) and two unipinnate setae; distal endite with one spine (bearing few spinules) and two setae (bearing several spinules medially). Allobasis drawn out into a stout claw (bearing few spinules subdistally) accompanied by three setae proximally. Endopod represented by four setae fused together basally.

Maxilliped (Fig. 10E) prehensile, three-segmented. Syncoxa elongate, about 3.2 times as long as wide, with one plumose seta subdistally. Basis elongate, 3.4 times as long as maximum width, unarmed. Endopod small, one-segmented, with one curved, geniculate claw bearing spinular row.

P1 (Fig. 10F). Coxa wide, unornamented. Intercoxal sclerite wide. Basis smaller than coxa, with one large anterior pore, and one long plumose inner seta; without outer seta. Segmentation, setation, and ornamentation of both rami as in S. marcida sp. nov. Length ratio of EXP1–EXP3 1: 0.84: 0.84; outer spines on EXP3 unipinnate or pinnate, and distal setae unispinulose medially and uniplumose subdistally. Endopod not exceeding distal end of EXP3; ENP1 reaching middle of EXP2, with one uniserrate inner seta; inner margin convex medially; ENP2 narrower and 0.9 times longer than preceding segment; with two unispinulose and geniculate distal setae, of which inner one uniplumose proximally, and one small bare posterior seta subdistally.

Segmentation and setation of P2–P4 as in S. marcida sp. nov.

P2 (Fig. 11A). Praecoxa small, triangular. Intercoxal sclerite subrectangular, with concave distal margin. Coxa large, unornamented. Basis with one bare outer seta; anterior surface with one pore and one row of spinules. Length ratio of EXP1–EXP3 1: 0.66: 1.05; EXP1 and EXP2 with outer spinules subdistally and with distal surface frill, and spinous process bearing anterior spinules distally; EXP3 with spinular ornamentation along outer and distal margins; two distal setae on EXP3 bipinnate (ornamented with more distinct spinules than those of outer spines), of which inner distal seta 1.5 times longer than outer distal seta. Endopod one-segmented, reaching distal fourth of EXP1, with one outer row of fine spinules; with one distal spine, one small inner seta, and one uniserrate inner seta; distal spine ornamented with more distinct spinules and uniserrate inner seta more deeply serrate than that of S. marcida sp. nov.

P3 (Fig. 11B). Praecoxa unornamented. Intercoxal sclerite smaller than in P2. Coxa with two rows of minute anterior spinules. Basis with long plumose outer seta exceeding distal end of EXP1; anteriorly with one pore and one row of spinules as in P2. Length ratio of EXP1–EXP3 1: 0.66: 1.10; two distal elements of EXP3 ornamented with more distinct spinules than those of outer elements as in P2; inner setal armature of EXP3 uniserrate in distal fourth. Endopod one-segmented, reaching distal fourth of EXP1, with one stout pinnate spine distally and one small inner seta subdistally; distal margin with hyaline frill.

P4 (Fig. 11C). Praecoxa and coxa larger than those of P2 and P3; coxa with two rows of posterior spinules. Intercoxal sclerite smaller than in P2 and P3. Basis with one bare outer seta; anteriorly with one pore and one row of spinules as in P2 and P3. Exopod distinctly longer than in P2 and P3; length ratio of EXP1–EXP3 1: 1: 0.76; EXP1 and EXP2 each with three groups and one group of outer spinules; distal margin with hyaline frill and expanded roundly; ornamentation of EXP2 as in EXP1 except for presence of only one group of outer spinules; two outer spines of EXP3 stouter than those of S. marcida sp. nov., inner distal seta 1.3 times longer than outer distal seta. Endopod as long as EXP1, two-segmented; ENP1 and ENP2 each with hyaline frill distally; ENP1 unarmed, and ENP2 with one fishbone-like spine distally and one small posterior seta subdistally.

P5 (Fig. 11D). Baseoendopod and exopod fused into a robust plate, with three anterior secretory pores proximally; with eight elements: seta a spiniform, stout, fused to segment basally, reaching proximal quarter of next strong spine (seta b), with strongly serrate tip; seta b also spiniform, 1.5 times as long as ramus; setae c–g setiform, seta c shorter than seta b, and seta d 1.8 times as long as ramus; setae e and f pinnate subdistally, seta e exceeding tip of seta b, twice as long as seta f; seta g bare, slightly exceeding tip of seta a; outer basal seta h longest, bi-articulate basally, and uniplumose subdistally.

Description of male (based on NIBRIV0000900840). Body length 912.9 µm (range = 859.9–978.3 µm; mean = 934.6 µm; n = 5; Korean population); habitus (Fig. 8C) as in female, but sexual dimorphism expressed as follows.

Urosome (Figs. 8C and 12A) six-segmented; genital and first abdominal somites separate.

Caudal rami (Figs. 12A and 12B) about 2.5 times longer than maximum width, tapering towards posterior end; with seven setae; position of each seta as in female; seta I diminutive, shortest as in female; seta II slightly longer than ramus; seta III 1.3 times as long as ramus; seta IV slightly shorter than seta II; principal seta V elongate, composite, composed of slender spiniform proximal part and long setiform distal part, as long as urosomites 1–4 combined.

Antennule (Fig. 9C) haplocer, ten-segmented, with geniculation between seventh and eighth segments. First segment small, with one bare seta and two rows of posterior spinules. Second segment longest, about 3.6 times as long as maximum width, with two plumose and seven bare setae, and one pore. Third segment tapering distally, with seven setae. Fourth segment minute, with two setae distally. Fifth segment swollen, with seven setae, one cylindrical process bearing one slender seta and one long aesthetasc (fused basally). Sixth segment small, with one small and one long seta. Seventh segment elongate, about twice as long as wide, with two setae. Eighth segment shorter than preceding one, with one small seta. Ninth segment 0.36 times as long as preceding one, unarmed. Terminal segment 3.5 times as long as preceding one, tapering distally, with one bare and six bi-articulate setae, and acrothek. Armature formula: 1-[1], 2-[9], 3-[7], 4-[2], 5-[5 + (1 + ae)], 6-[2], 7-[2], 8-[1], 9-[0], 10-[7 + acro].

P2 (Fig. 12C). Protopod and intercoxal sclerite as in female. Inner distal corner of basis forming a distinct spinous process anteriorly. Exopod three-segmented; EXP1 and EXP2 as in female, but outer distal corner less produced and ornamented with small anterior spinules; outer spines on each exopodal segment smaller than those in female; EXP3 longer than that in female, as long as EXP1 and EXP2 combined; inner margin wavy, with two groups of setules; distal element modified into a strong spine recurved distally and with one row of spinules medially and one minute setule apically; with three pinnate outer spines. Endopod two-segmented, as long as EXP1; ENP1 with one serrated seta posteriorly; ENP2 slightly shorter than preceding segment, with one apical seta curved outwardly.

P3 (Fig. 12D). Praecoxa (not figured) and coxa unornamented. Intercoxal sclerite (not figured) as in female. Basis with long plumose seta. Exopod as in female, but EXP3 relatively shorter than in female. Endopod two-segmented; ENP1 small, unarmed; ENP2 strongly modified into a recurved apophysis, about 3.9 times as long as preceding segment, with one unipinnate distal seta.

P5 (Figs. 12A and 12E) as in S. marcida sp. nov.; with two anterior pores and six setae: seta a shortest; seta b twice as long as ramus; seta c 0.6 times as long as seta b; seta d short, twice as long as seta a; seta e slightly shorter than seta d; and outer basal seta f longest, uniplumose subdistally.

P6 (Figs. 12A and 12F) as in S. marcida sp. nov.; with two asymmetrical plates, only one leg functional, each with three setae, of which outer one about two times longer than others.

Variability.—The length ratio of seta b of the female P5 is variable between populations. The length ratio of seta b to the ramus of P5 in the Korean population is higher (1.6 times) than in the Japanese (see Itô, 1972: 328, Figs. 15-5 and 15-6) and Russian (see Fig. 11E) populations (about 0.9 times). It is likely that this morphological variability depends on the body size of the organisms of each population; The specimens of the Korean population (966.1–1,087.3 µm) are smaller than the specimens of the two other populations (1,200.0 µm in the Japanese population; 1,077.3–1,245.5 µm in the Russian population).

Remarks.—This marine harpacticoid was originally described by Itô (1972) based on both sexes living on an intertidal sandy beach at Akkeshi, Hokkaido, Japan, and subsequently recorded in sublittoral environments in the White Sea (Russia) by Kornev & Chertoprud (2008). The latter authors provided a brief description of the body and caudal rami, the armature of P2–P4, and a drawing of the female’s caudal rami, recognizing only differences in the length of female (1,200 μm in the Japanese specimens vs. 923 μm in the Russian specimens). However, Kornev & Chertoprud’s (2008) setal armature differs from the original description. Itô (1972) considered a setule-like element near the inner subdistal margin in P2–P3 ENP1 and P4 ENP2 as a setule. However, based on both the illustrations of Itô (1972: Fig. 15) and our observation of specimens from the East Sea (arrowheads in Figs. 11A–11C), this element can be interpreted as a minute seta, indicating that the armature patterns of P2–P4 endopods are ‘210’, ‘110’, and ‘0.110’, respectively. The fact that Kornev & Chertoprud (2008) provided the armature formula of P4 endopod as ‘0.010’ conceivably implies either that their report is an observational error or that the specimens from the White Sea actually represent a new and as yet undescribed distinct species with close affinity to S. intermedia. Without re-examination of Kornev & Chertoprud’s specimens, the latter decision might be premature because the complete description of the armature complement of P4 was omitted in older records of Stenocaris species (cf. Scott, 1892, 1900; Sars, 1909).

Stenocaris intermedia displays certain features in common with S. gracilis, i.e., the sexual dimorphic condition of male P3 endopod consisting of ENP1 without any apophysis and ENP2 modified into a long apophysis, the female six-segmented antennule, and female P5 with a stout seta a (fused to ramus) and a spiniform seta b. However, these two species can be distinguished based on three features: the relative length of the terminal to the penultimate segments in the female antennule (at least threefold in S. intermedia vs. twofold in S. gracilis); the element on male P3 ENP2 (delicate and uniplumose in S. intermedia vs. stout, recurved, and spine-like in S. gracilis); and the structure of female caudal seta V (distinctly small and stout spine-like in S. intermedia vs. non-composite seta-like in S. gracilis; approximately one-third the length of the caudal ramus in S. intermedia vs. extremely long, about 5 times the length of the caudal ramus in S. gracilis) (Sars, 1909; Itô, 1972). The latter characteristic is the most conspicuous feature of S. intermedia.

Our specimens of S. intermedia from the East Sea (South Korean and Russian waters) largely correspond to the original description. However, we observed several minor differences that can be considered intraspecific variability: the mandibular palp is more elongated and the two outer elements on P4 EXP3 are more developed in our specimens; also, seta b on the female P5 is distinctly longer than that of Itô’s (1972) specimen (about threefold as long as seta a in our specimens vs. twofold in the Japanese specimens). When Itô (1972) described this species, he presented two morphological variations in the female P5 in the length-to-width ratio of a single segment and the thickness and length of setae/spines. We observed similar variations between specimens from the same locality or geographical population (South Korean vs. Russian specimens). Thus, we assumed that S. intermedia has high intra-population variability in this leg, although the details of P5 are important for separating harpacticoid species (e.g., Kim, Lee & Huys, 2021; Yeom et al., 2021).

Discussion

On the taxonomic status of the Stenocaris minor complex

Based on only female specimens, Scott (1892) originally described this species (as Cylindropsyllus minor at that time) from the Firth of Forth, Scotland, with a brief description and illustrations of the body, antennule, maxilliped, and P1–P5 (his figures of P2 and P3 were erroneously labeled P3 and P2, respectively; see Scott, 1892: Plate XL, Figs. 21, 22). Subsequently, he revised the previous description of the female specimen (in particular, the relative length ratio of antennular segments, and setation of both rami of P1) and provided information on the males including the sexual dimorphic features of P2 and P3, based on specimens from the type locality (see Scott, 1900: Plate XIV, Figs. 23–32). Since this correction, S. minor has been reported from a variety of coastal regions: Korshamn, Norway (Sars, 1911); Helgoland Island, Germany (Kunz, 1938); Roscoff, France (Monard, 1935); Naushon Island, and Massachusetts (Wilson, 1932); the Bulgarian coast of the Black Sea (Apostolov, 1971; Marinov, 1971; Apostolov & Marinov, 1988); and Figarolo Island (Olbia, Italy) in the Mediterranean Sea (Cottarelli & Venanzetti, 1989).

The description of Sars (1911) has been accepted as an appropriate standard for Stenocaris minor (Kunz, 1938; Lang, 1948). However, the fact that most of subsequent records include insufficient descriptions and illustrations, or are occasionally known from only one sex, may have hampered comprehensive comparisons, possibly broadening the taxonomic concept of S. minor. Because harpacticoid species complexes can be divided into several cryptic species (e.g., Fiers & Kotwicki, 2013; Huys & Mu, 2021; Karaytuğ et al., 2021), some of these records might be part of a Stenocaris minor complex. Monard (1935) and Kunz (1938) reported the characteristics of males of this species from European waters near the type locality. Despite discrepancies in details of P2 (length ratio of segments and setal armature), they were considered identical to Stenocaris minor due to their geographical proximity. Several authors have documented the sympatric occurrence of Stenocaris minor from the Black Sea, but the descriptions given by Apostolov (1971), Marinov (1971), and Apostolov & Marinov (1988) deviate from S. minor s. str. in several respects. Apostolov’s (1971) specimens have only one outer spine on P3–P4 EXP3, no inner seta on P4 EXP3, and female P5 is broad and has a distinct notch on the distal margin, conceivably reminiscent of the legs of immature copepodid stages or that of different species (see Apostolov, 1971: Figs. 29–31). Except for observable errors in male P2 ENP2 without a distal element and the reduced setation of male P5, the Bulgarian specimens of Marinov (1971) differ from the Scottish specimens described by Scott (1900) in the sexual dimorphism of male P3 endopod (one-segmented in the Bulgarian specimens vs. two-segmented in the Scottish specimens and other records), raising doubts about its actual identity. The caudal rami of S. minor sensu Apostolov & Marinov, 1988 were illustrated as possessing composite terminal seta V (with the setiform part distally), at least 1.5-fold as long as the caudal ramus (vs. slightly longer). Despite these differences, limited information and inaccuracy have hampered determination of the Black Sea specimens’ taxonomic identities. Pending collection of Stenocaris species from the Black Sea, S. minor sensu Apostolov, 1971, S. minor sensu Marinov, 1971, and S. minor sensu Apostolov & Marinov, 1988 are here relegated to species inquirenda of Stenocaris.

Wilson (1932) reported the first occurrence of S. minor in North America based on females collected from freshwater in Naushon Island, Massachusetts. His insufficient description and figures of the caudal ramus and female P5 hamper morphological comparison between American specimens and the reports of Scott (1900) and Sars (1911). Re-description of the species is essential to rule its identity and would facilitate the discovery of a new species given the wide geographical area. Therefore, the American population is also regarded as a species inquirenda in the genus.

Stenocaris figaroloensis sp. nov.

urn:lsid:zoobank.org:act: AE4F4E54-05E9-4F05-8AD6-55E8CE752934

Stenocaris minor (T. Scott, 1892) sensu Cottarelli & Venanzetti, 1989

Original description.—Cottarelli & Venanzetti (1989—as Stenocaris minor): 202–204; Fig. 10.

Type locality.— Italy, Figarolo Island (the Tyrrhenian Sea); sandy sediments at a depth of 1 m.

Type material.—The female and male specimens illustrated by Cottarelli & Venanzetti (1989) in his Fig. 10 are here fixed as the syntypes of S. figaroloensis sp. nov. in accordance with International Commission on Zoological Nomenclature (1999) Arts. 16.4, 72.3, and 72.5.6.

Etymology.—The species name is named after the type locality of the new species. It is in the nominative singular, gender feminine.

Remarks.—Cottarelli & Venanzetti (1989) recorded both sexes of Stenocaris species under the name S. minor from the Tyrrhenian Sea, the Mediterranean Sea (Italy), providing details of the mandible, P2–P5, and male P6. Although their description contained some uncertain characteristics, such as the absence of a setule-like element on P2 ENP1 and P4 ENP2, only one outer spine on P4 EXP3, and male P3 endopod comprising only one segment, the Italian population differs significantly from the re-descriptions of S. minor by Scott (1900) and Sars (1911): (1) The mandibular endopod has one lateral and three apical setae (vs. one lateral and four apical setae). (2) Male P2 ENP1 lacks a long seta posteriorly (vs. present). (3) The relative length and nature of elements on the female P5, i.e., the stout seta a is remarkably uniserrate medially and flagellate distally (this form is not known from other congeners), and setae c, f, and g are markedly shorter than other elements. Based on these differences, we attribute Cottarelli & Venanzetti’s (1989) specimens to a distinct species named S. figaroloensis sp. nov.

On the polyphyletic status of Stenocaris

The monophyly of the genus Stenocaris has been questioned in many previous works (e.g., Apostolov, 1982; Huys & Conroy-Dalton, 1993; Kunz, 1994; Moura & Pottek, 1998; Kornev & Chertoprud, 2008). This taxonomic confusion might have been resulted from the long-accepted simple criteria used to distinguish cylindropsyllid genera, such as segmentation of the thoracic legs and the degree of development of the maxilliped. However, recent researches have clarified the characteristics of the body and caudal rami, the nature of the mouth appendages, the sexually dimorphic structures of P2–P3, and the setation pattern of the swimming legs as well as P5 in both sexes to (re-)define the general classification of the family Cylindropsyllidae (Huys & Conroy-Dalton, 1993; Moura & Pottek, 1998; Huys & Conroy-Dalton, 2006a, 2006b; Kornev & Chertoprud, 2008; Richter, 2019). In addition, Richter (2019) recognized that any apomorphies for the genus have not been defined. Following these approaches, a reevaluation of the morphological features of the species of Stenocaris is needed to support the monophyly of the genus.

Arlt (1983) originally described Stenocaris baltica based on only one male specimen collected from a fine sandy substrate at 13-m depth in the Baltic Sea. Although he claimed that a relationship of this species with Stenocaris was supported by the well-developed maxilliped (being not reduced as in Cylindropsyllus) and the six-segmented antennule, his illustrations of the antenna and swimming legs (Arlt, 1983: Fig. 14) raised doubts about the true generic identity of S. baltica since it could be attributed to Vermicaris. The following characteristics of this species appear to be strongly related to the latter genus: the total body length (440 μm in the male) is distinctly smaller than other Stenocaris species (about 1,000 μm); the antennary exopod is extremely reduced and expressed as a small seta (vs. one-segmented, with two apical setae in Stenocaris), P1 ENP2 has only two apical setae (vs. three setae in Stenocaris), male P2 ENP2 is a semicircular process lacking a distal element (vs. oblong, bearing one distal seta in Stenocaris), and male P3 ENP2 is one-segmented (vs. two-segmented in Stenocaris). The P5 in both sexes of Cylindropsyllidae is a single plate, but Arlt (1983) questionably mentioned that the exopod is separated from the baseoendopod. Note that Arlt’s (1983: Fig. 14) illustration of P5 bearing a single apical seta is probably reminiscent of male P6, and he stated that the baseoendopod was damaged during preparation. This is undoubtedly based on an observational error. Given these morphological features, this species should be reinstated in Vermicaris as V. baltica (Arlt, 1983) comb. nov.

Noodt (1955) described Stenocaris pygmaea from the Bay of Biscay in the Atlantic Ocean based on one female specimen (body length 370 μm), stating its close affinity with both S. minuta (= V. minuta) and S. pontica (= V. pontica). Based on the morphology of the Bulgarian species identified as S. pontica, Marinov (1971) presumed this species to be identical to S. pontica described from the Romanian coast of the Black Sea, and subsequent authors (e.g., Apostolov, 1972; Apostolov & Marinov, 1988) considered it as a junior synonym of the latter. However, S. pygmaea was reinstated as a valid species by Huys & Conroy-Dalton (1993), who pointed out a significant difference in segmentation of the P2 endopod between the two species, supporting the removal of its synonym with S. pontica. Although this unusual condition of S. pygmaea deviates from the generic boundary of Vermicaris, this species can be allocated to the latter as V. pygmaea (Noodt, 1955) comb. nov. rather than being placed in Stenocaris. This is substantiated in the one-segmented female P3 endopod, semicircular apically, and lacking any elements, and in that female P5, which is smaller than those of other genera, with reduced armature with 2–3 setae. We suggest that these two characters could be potential autapomorphies for Vermicaris.

Within the family, the morphology of caudal rami in S. arenicola and S. kliei displays a reduced ramal length and a dorsal spinous process. Members of the family, except for Monsmeteoris Richter, 2019, have elongated caudal rami that are at least twice as long as broad. In the Navalonia clade comprising the genera Bolbotelos, Boreovermis, Willemsia, and Navalonia, it is modified into bulbous appendages (Huys & Conroy-Dalton, 1993). However, in S. arenicola and S. kliei, these rami are strongly reduced: they are as long as broad and have a convex inner margin. Huys & Conroy-Dalton (1993) proposed phylogenetic relationships among Boreopontia heipi Willems, 1981, S. arenicola, and S. kliei based on the presence of a dorsal spinous process on the caudal rami; they also re-examined Wilson’s collection of S. arenicola, for which Wilson (1932) gave only a brief drawing of the habitus. Although four Stenocaropsis species have a process similar to the above three species, there are also fundamental differences (Huys & Conroy-Dalton, 1993: 293). Huys & Conroy-Dalton (1993) suggested that this recurved process is derived from a dorsal integumental extension, which can be interpreted as convergent evolution in B. heipi and the S. arenicola/S. kliei group based on the significant differences in the sexual dimorphism of P2 and P5. Indeed, several fundamental discrepancies between them can be readily noticed. In male P2–P3, the exopodal segments are strongly modified in the S. arenicola/S. kliei group as in other genera (see the remarks on Huysicaris gen. nov.), whereas B. heipi does not exhibit such sexual dimorphism. Male P5 of B. heipi is characterized by the modification of seta a or b into a spine bearing a crenate tip and a distal setule, of seta c or d into a stout spine bearing a distal setule in females, and of seta b or c (second innermost) into a stout spine in males (see Huys & Conroy-Dalton, 2006b: Figs. 36–37). In the insufficient original illustration of S. arenicola by Wilson (1932), male P5 exhibits seven setae, which is the maximum number of setae and is known only in Evansula T. Scott, 1906, and seta a (innermost) is modified into a stout spine. A similar modification in male P5 was present in the original description of S. kliei, although seta a is fused to the segment basally (see Kunz, 1938: Abb. 12, Fig. 8). Indeed, two different types of modified spines are present in the same genus, i.e., it is defined in the female of S. minor s. str. and S. figaroloensis sp. nov. vs. fused in S. gracilis, S. intermedia, and S. marcida sp. nov. The homologous position of the spine in S. arenicola and S. kliei could support Huys & Conroy-Dalton’s (1993) review and their isolated taxonomic position within Stenocaris. Both species can be distinguished from Stenocaris in the extreme elongation of the penultimate somite as in Boreopontia and the Navalonia clade; the lack of caudal seta I; the second antennulary segment being not elongated in both sexes; P1 ENP2 with only two (sub)distal setae; and both males P5 separate (vs. medially confluent in Stenocaris; see below). Considering these characteristics and the generic discriminant, we propose a new genus Huysicaris gen. nov. to accommodate H. arenicola (Wilson, 1932) comb. nov. and H. kliei (Kunz, 1938) comb. nov. to move toward the monophyly of Stenocaris.

Genus Huysicaris gen. nov.

urn:lsid:zoobank.org:act:18ADF050-EF12-4198-86EA-1DE3C2BA1C83

Type species.—Huysicaris kliei (Kunz, 1938) comb. nov. [by original designation]

Other species.—H. arenicola (Wilson, 1932) comb. nov.

Diagnosis.—Cylindropsyllidae. Body vermiform, without separation between prosome and urosome; penultimate somite extremely elongated; anal operculum wide. Rostrum triangular, defined at base. Caudal rami as long as broad or slightly longer, with a recurved spinous process on dorsal margin; inner margin bulbous; with six setae, seta I lost. Antennule elongate, 7-segmented in ♀, with aesthetasc on segments 4 and 7; haplocer and 8(?)-segmented in ♂. Antenna with allobasis; exopod small, 1-segmented, with 2 distal setae. Mandible, maxillule, and maxilla unknown. Maxillipedal syncoxa with 1 subdistal seta; endopod represented by claw, with or without an accompanying small seta. P1 exopod 3-segmented; EXP2 and EXP3 lacking inner element. P1 endopod 2-segmented; ENP1 with 1 inner seta, ENP2 with 2 distal setae. P2 basis with hook-like process anteriorly. P2–P4 exopods 3-segmented; P4 exopod extremely elongated, P4 EXP3 shortest; P2–P3 exopods strongly modified (stout and bending inwardly) in ♂; inner distal seta on P2 EXP3 modified into a claw in ♂. P2–P3 endopods 1-segmented and P4 endopod 2-segmented; P3 ENP1 with long and stout apophysis in ♂. Setal armature formula of P1–P4 as in Table 2.

Table 2 Armature formula of P1–P4 in the genus Huysicaris gen. nov.

	Exopod	Endopod	
P1	0.0.021–2	1.011	
P2	0.0.022 [0.0.011–2 in ♂]	110 [01–20 in ♂]	
P3	0.0.122 [0.0.012 in ♂]	0.010 [apo.?]	
P4	0.0.0–121–2	0.0–110	

P5 forming single plate, with 1 spinose process and 6(?) setae in ♀; in ♂ with very long basal seta and 5–6 setae, of which the innermost seta modified into stout spine (fused to plate basally in H. kliei comb. nov.).

P6 ♀ unknown; in ♂, represented by opercula, each bearing 1 short and 2 long setae.

Etymology.—The generic name is dedicated to Dr. Rony Huys (National History Museum, London), in recognition of his numerous contributions to harpacticoid diversity and taxonomy including the family Cylindropsyllidae. It is formed by Dr. Rony Huys’ family name and the third declension of the Greek feminine noun cāridis, cāris, sea-crab, or shrimp. It is in the nominative singular, gender feminine.

Notes.—Due to incomplete original descriptions of H. arenicola comb. nov. and H. kliei comb. nov., some characters of the generic diagnosis remain undefined.

Remarks.—Huys & Conroy-Dalton (1993) pointed out the different sexual dimorphism in P2 and P3 of H. arenicola comb. nov. and H. kliei comb. nov. Incomplete descriptions and vague illustrations by Wilson (1932) and Kunz (1938) hamper detection of (aut)apomorphies for this new genus and determination of its taxonomic position. However, in accordance with Kunz’s (1938) figures, male P3 of H. kliei comb. nov. displays two uncommon characteristics: (1) EXP3 is recurved subdistally and inwardly, and has only one distal and two outer elements; and (2) the endopod is probably three-segmented, with a stout recurved apophysis on ENP2 reaching the distal end of EXP3. The former is regarded here as an autapomorphy for Huysicaris gen. nov. because it is not expressed in any cylindropsyllid genera; unfortunately, Wilson (1932) did not provide information on male P3 of H. arenicola comb. nov. In the family, the three-segmented condition of male P3 endopod is retained in Evansula, which occupies a basal position in the family (cf. Huys & Conroy-Dalton, 2006b). Although Huysicaris gen. nov. might retain this plesiomorphic condition, the presence of an unusual spinous process on male P2 basis in the both species (see Wilson, 1932: Plate 17f; Kunz, 1938: Abb. 12, Fig. 6) strongly indicates the new genus has a close relationship with all other cylindropsyllid genera (except for Evansula) representing this significant characteristic as a synapomorphy.

Another taxonomic problem with Stenocaris is that autapomorphies have not been defined. Removal of four species—H. arenicola comb. nov., H. kliei comb. nov., V. baltica comb. nov., V. pygmaea comb. nov.—from Stenocaris might justify its generic concept and determine its autapomorphies within the Cylindropsyllidae. The sexual dimorphic features of male P2 and P3 are the primary characters used to reconstruct the phylogenetic relationships among cylindropsyllid genera, i.e., the presence of a minute or recurved apophysis on P2 basis, a modified distal spine on P2 EXP3, fusion of P2 EXP2 and EXP3, and a recurved or minute apophysis on male P3 endopod (Moura & Pottek, 1998; Huys & Conroy-Dalton, 2006b). Based on the latter feature and the one-segmented endopods in female P2–P3, Moura & Pottek (1998) presumed that Stenocaris is nested in the Navalonia clade. However, Huys & Conroy-Dalton (2006a) demonstrated that the Navalonia clade can be characterized by allometric growth of male P3 endopod. Within the genus, males of S. gracilis, S. intermedia, and S. marcida sp. nov. lack an apophysis on male P3 ENP1, although it is present in both S. minor s. str. and S. figaroloensis sp. nov. Despite this discrepancy, the fifth legs of five Stenocaris species (S. gracilis, S. intermedia, S. marcida sp. nov., S. minor s. str. and S. figaroloensis sp. nov.) are fused medially (vs. separated in all other genera; cf. Huys & Conroy-Dalton, 2006b: Fig. 37), and it is regarded as the most distinct autapomorphy for Stenocaris.

Conclusions

We describe a new marine harpacticoid, Stenocaris marcida sp. nov., from the subtidal sediments off Dok-do Island in the East Sea of South Korea, and report on the occurrence of S. intermedia Itô, 1972 based on specimens collected from subtidal sediments of Ulleung-do Island in South Korean waters and from the coast of Primorsky in Russian waters. Our detailed morphological comparison of all Stenocaris species and their taxonomic records provides insight into the generic concept and confirms that the nature of the fifth leg, caudal rami, and sexual dimorphic features of the second and third legs can be used to define the generic boundary of Stenocairs. The monophyly of the genus is supported by a significant synapomorphy, the confluent condition of the fifth legs in males, and the genus is comprised of S. gracilis Sars, 1909, S. minor (T. Scott, 1892), S. intermedia, S. figaroloensis sp. nov., and S. marcida sp. nov. Based on this taxonomic review, other Stenocaris species were transferred to the genus Vermicaris Kornev & Chertoprud, 2008 or to the new genus Huysicaris gen. nov. as V. baltica (Arlt, 1983) comb. nov., V. pygmaea (Noodt, 1955) comb. nov., H. arenicola (Wilson, 1932) comb. nov., and H. kliei (Kunz, 1938) comb. nov. Further studies on the S. minor species complex will enhance understanding of the diversity of cryptic species and the phylogenic relationships among cylindropsyllid harpacticoids.

The authors would like to thank Dr. Sung Hoon Kim (the Korea Polar Research Institute, Incheon, South Korea) and Mr. Joo Won Kang (Chosun University, Gwangju, South Korea), and Dr. Tae Won Jung (the Honam National Institute of Biological Resources, Mokpo, South Korea) for their help in collecting sediment samples from the Russian waters and Ulleung-do Island in South Korea. We are also grateful to the anonymous reviewers for their critical comments and help in improving the manuscript.

Additional Information and Declarations

Competing Interests

Author Contributions

Field Study Permissions

Data Availability

New Species Registration

The authors declare that they have no competing interests.

Jong Guk Kim conceived and designed the experiments, performed the experiments, analyzed the data, prepared figures and/or tables, authored or reviewed drafts of the article, and approved the final draft.

Kyuhee Cho performed the experiments, analyzed the data, prepared figures and/or tables, and approved the final draft.

Seong Myeong Yoon conceived and designed the experiments, analyzed the data, authored or reviewed drafts of the article, and approved the final draft.

Jimin Lee conceived and designed the experiments, performed the experiments, analyzed the data, authored or reviewed drafts of the article, and approved the final draft.

The following information was supplied relating to field study approvals (i.e., approving body and any reference numbers):

Field experiments were approved by the National Marine Biodiveristy Institue of Korea (project number: 20170431), the National Institute of Biological Resources (project number: NIBR201501201).

The following information was supplied regarding data availability:

The type material of Stenocaris marcida sp. nov. is reposited in the National Marine Biodiversity Institute of Korea (MABIK; MABIK CR00252790, MABIK CR00252791, MABIK CR00252792, MABIK CR00252793) and the examined specimens of Stenocaris intermedia are deposited in the National Institute of Biological Resources (NIBR; NIBRIV0000900838, NIBRIV0000900839, NIBRIV0000900840, NIBRIV0000900841, NIBRIV0000900842, NIBRIV0000900843, NIBRIV0000900844).

The following information was supplied regarding the registration of a newly described species:

Publication LSID: urn:lsid:zoobank.org:pub:9F8C21FE-74C1-4F7C-BA5C-22B9CC0869DB

Stenocaris marcida LSID: urn:lsid:zoobank.org:act:45861574-3D8E-42CB-ADEE-E32EA4F3334D

Stenocaris figaroloensis LSID: urn:lsid:zoobank.org:act:AE4F4E54-05E9-4F05-8AD6-55E8CE752934

Huysicaris LSID: urn:lsid:zoobank.org:act:18ADF050-EF12-4198-86EA-1DE3C2BA1C83

Stenocaris LSID: urn:lsid:zoobank.org:act:FA34B215-A8B2-4A3D-B8E3-B958019A1DDD.

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
