# Peer review of "Taxonomic review of the genus Stenocaris Sars (Copepoda, Harpacticoida, Cylindropsyllidae), with (re)descriptions of two Stenocaris species from the Far East"

_PeerJ, doi:10.7717/peerj.14623_

## Round 0.1 · original submission · Minor Revisions

All reviewers applaud your research and writing. However, there are several parts in your manuscript that need modifications before it can be accepted for publication, which I think is minimal.

·

Basic reporting

The manuscript is well-written and structured. The authors' style is unambiguous and professional, but I took the liberty to suggest some corrections to improve the readability of the text. The authors cited the most relevant literature. The figures and tables are very informative, and the figures are of excellent quality. The results and conclusions are well-supported.
On the other hand, I suggest some corrections to completely adhere to the provisions of the International Code of Zoological Nomenclature.

Experimental design

The content of the manuscript fits the scope of the journal. The research questions and well-justified and defined. The methods are of high technical levels and are described with sufficient detail.

Validity of the findings

This manuscript is a nice piece of research with meaningful findings for the field. The information given in the manuscript is robust and is based on sound analyses of the available literature. The conclusions are sound and linked to the original research question.

Additional comments

This is a nice manuscript with the description of a new species of the genus Stenocaris. The authors' contribution deals with the re-diagnosis of the genus (they restricted the genus to five species, and other species were removed to another new genera proposed by the authors, and to other genera erected previously). The text is well written and structured, but I suggested some minor corrections to improve the readability of the text. Additionally, I suggested some corrections to adhere to the provisions of the ICZN. The figures are very informative and of excellent quality.

Reviewer 2 ·

Basic reporting

Overall, the manuscript is ready and acceptable in quality for publication. The abstract statement is clear and reflect the outcome of the study. The introduction, literature review and methodology are appropriate and satisfactory by covering relevant detail with good use of citation. For the result and discussion, the author able to integrate the data/findings with current and critical points of literatures. Only several corrections were pointed out in the manuscript.

Experimental design

The author(s) clearly defined and raised the key issues of systematics on this genus. The approach of these issues are appropriate and sufficient to answer and fill the knowledge gap. Nevertheless, perhaps the author could explain further on mounting process of the specimens. Furthermore, the author also should explain on the cladistic analysis as it is discuss in the manuscript.

Validity of the findings

This study is highly interesting, highly novel and on a highly important question.

Annotated reviews are not available for download in order to protect the identity of reviewers who chose to remain anonymous.

Reviewer 3 ·

Basic reporting

No comment

Experimental design

No comment

Validity of the findings

No comment

Additional comments

This manuscript focusing on the genus Stenocaris Sars with (re)descriptions of Stenocaris marcida and Stenocaris intermedia from the Far East. Despite the significant characteristics S. marcida sp. nov. shared with S. minor s. str. , there are 6 conspicuous different between those two species. Meanwhile, although S. intermedia share certain features with S. gracilis, they can be distinguished based on 3 features, especially the structure of female caudal seta. Meticulous detailed illustrations.

Introduction is well prepared.

This reviewer would like to add a few comments which might strengthen the presentation and discussion of Kim et al. findings:

L33: ‘the sexual dimorphism of the second leg,’ Female or male? Please clarify.

L67: misspelled; change Richer to Richter

L124: ‘10% formalin solution’. Did the formaldehyde solution from the lab itself comes as 37%? If yes, the formalin solution used should be 4%.

L170-171: ‘Antennule six- or seven-segmented,’. Please be precise. It looks like seven-segmented.
L187: Fig. 5 (A), S. marcida sp. nov. Please confirm setal armature for endopod P2, P3

Discussion is thoughtfully presented.

Annotated reviews are not available for download in order to protect the identity of reviewers who chose to remain anonymous.

---

## Round 0.2 · Minor Revisions

While the manuscript has been improved significantly following the first reviewing process. However, there are several areas in the manuscript that need to be improved particularly the language. I would also suggest (though not compulsory) sending this manuscript to a proofreading service (which is also offered by PeerJ) for a final language check and correction.

·

Basic reporting

As for the first round of revisions.

Experimental design

As for the first round of revisions.

Validity of the findings

As for the first round of revisions.

Additional comments

This is the second round of revisions of this manuscript. The authors did a good job and addressed all my comments and suggestions. I only found some minor mistakes (mostly slips of the pen). This manuscript deserves to be published with some minor corrections. There is no need for a third round of revisions.

---

## Round 0.3 · accepted · Accept

All comments by the reviewers have been addressed, including the language. Thus, I think the manuscript is now ready for the next step of publication.